# Generating high fidelity wind fields from the wind speed correlation tensor

Matteo Faccioni 1,2, Daniel Kiehn3, and Patrick Vrancken1

Correspondence: Matteo Faccioni (matteo.faccioni@dlr.de)

**Abstract.** In this publication a new method to generate stochastic representations of homogeneous and isotropic wind fields is presented. In contrast to the typically employed algorithm, the new approach is based on the wind speed correlation tensor. This allows simulating a homogeneous and isotropic turbulent wind field with very high accuracy. In this publication, a deviation of the obtained dataset's structure function from the theoretical one of at least one order of magnitude lower than the commonly used method is achieved. Furthermore, a compensation method to decrease this error even further is proposed. Moreover, being a generic method, it can be used to simulate other Gaussian phenomena (e.g., temperature or index of refraction fluctuations) on various spatial domains and grid shapes.

## 1 Introduction

In recent decades, researchers and engineers have increasingly relied on the use of synthetic datasets to represent complex physical phenomena during the design process. Examples include the generation of wind speed along the three spatial dimensions, used in the design of airframe structures (cf. (Hoblit, 1988) or the aviation regulation (CS25, 2023)) and the computation of wind turbine loads (as specified in (IEC61400-1, 2019)). Further, synthetic datasets are relevant for the generation of phase screens, used in the modeling of optical propagation through the turbulent atmosphere (Andrews and Phillips, 2005). For this reason, within the last sixty years, several stochastic synthesis methods have been developed, in particular for phenomena that are considered Gaussian and stationary (e.g., temperature fluctuations and homogeneous, isotropic turbulence). These methods are based on different statistical approaches, including modal decomposition techniques (Fried, 1965), autoregressive models (Baran and Infield, 1995), linear dynamical system solutions (Beghi et al., 2011), machine learning (Wold et al., 2024), or the spectral representation of the phenomenon. This last family of algorithms can be further divided into the methods computing the dataset by means of the relation between the phenomenon's correlation function and power spectral density (PSD) (Dietrich and Newsam, 1997; Borgman et al., 1984), and the ones synthesizing the dataset directly from the spectrum of the phenomenon (Mann, 1998). The latter class of algorithms, referred in this article as *random phase method* (RPM) after (Wilson, 1998), has been very widely used in the realization of stochastic wind fields in the last twenty years (Friedrich et al., 2022; Chen et al., 2022). However, to the authors' knowledge, there is still no algorithm capable of synthesizing a high-fidelity wind field dataset

<sup>&</sup>lt;sup>1</sup>DLR Institute of Atmospheric Physics, Oberpfaffenhofen, Germany

<sup>&</sup>lt;sup>2</sup>Technical University of Munich, School of Engineering and Design, Department of Aerospace and Geodesy, Institute of Astronomical and Physical Geodesy, Munich, Germany

<sup>&</sup>lt;sup>3</sup>DLR Institute of Flight Systems, Braunschweig, Germany

(i.e., achieving a deviation from the theoretical correlation of less than the 1%) on different spatial domains (e.g., a cuboid of  $2000 \,\mathrm{m} \times 2000 \,\mathrm{m} \times$ 

## 35 2 The legacy approach: Random Phase Method (RPM)

First applied by Shinozuka (Shinozuka and Jan, 1972) in the field of structural analysis, the approach to use the PSD in the synthesis of stochastic datasets has become basically *the* method of choice in different areas, such as wind engineering (Mann, 1998), acoustics (Wilson, 1998), and astronomy (Lane et al., 1992). Considering a stationary random field u(s), where s is the space vector and u is a vector composed of the components along the three axis  $u(s) = (u_x(s), u_y(s), u_z(s))$ , the field's spectral representation can be written in the form of a Fourier-Stieltjes integral with a random complex amplitude  $d\psi(k)$  (Cramer and Leadbetter, 1967):

$$u(s) = \int_{-\infty}^{+\infty} e^{i \, k s} d\psi(k); \qquad (1)$$

where k is the wavenumber vector, and i is the imaginary unit. The process  $\psi(k)$  is directly related with the PSD  $\Phi(k)$  by (Cramer and Leadbetter, 1967):

$$45 \quad \overline{|d\psi(\mathbf{k})^2|} = \Phi(\mathbf{k})d\mathbf{k}; \tag{2}$$

where the overline symbol  $\overline{\psi}$  denotes the mean. In the case of a three-dimensional turbulent wind field, the PSD is described by the  $3 \times 3$  tensor  $\Phi_{pq}(\mathbf{k})$  (Batchelor, 1953), where the subscripts p and q are the tensor indices. In a discretized volume, the integral in Eq. (1) can be approximated with a discrete Fourier series (Mann, 1998):

$$u_p(\mathbf{s}) = \sum_{i=1}^N e^{\mathrm{i} k_j \mathbf{s}} C_{pq}(\mathbf{k}) \mu_q(\mathbf{k}); \tag{3}$$

where N is the number of points in the dataset (i.e., the total number of the grid's pixels),  $\mu_q(\mathbf{k})$  is a set of complex random variables with zero mean and unit variance, and  $C_{pq}(\mathbf{k})$  are the Fourier coefficients that are related with the wind speed tensor

© Author(s) 2025. CC BY 4.0 License.

by (Mann, 1998):

$$C_{pt}^* C_{qt} = \frac{(2\pi)^m}{V} \Phi_{pq}(\mathbf{k}); \tag{4}$$

where V is the volume of the grid considered, the star symbol \* denotes the complex conjugate, and m is the number of dimensions of the considered spatial domain (e.g., m = 2 for a 2-D grid). Once computed the Fourier coefficients  $C_{pq}$ , Eq. (3) can be solved efficiently using the inverse Fourier transform (Cooley and Tukey, 1965):

$$u_p(\mathbf{s}) = \operatorname{Re}/\operatorname{Im}\left\{\mathcal{F}^{-1}\left(\mu(\mathbf{k})\frac{(2\pi)^{m/2}}{\sqrt{V}}N\sqrt{\Phi_{pq}(\mathbf{k})}\right)\right\};\tag{5}$$

where Re/Im symbol means that either the real or imaginary component of the result can be used,  $\mathcal{F}^{-1}$  stands for the Discrete Inverse Fourier Transform (DIFT). The  $\sqrt{\Phi_{pq}(\mathbf{k})}$  term can be computed using a matrix decomposition technique, such as Cholesky decomposition (Cholesky, 1910), or the dimension-dependent factorizations suggested in (Mann, 1998).

## 3 Verification of the synthesized dataset

To check the quality of the dataset generated using Eq. (5), i.e., computing the dataset's correlation deviation from the theoretical one, the dataset's structure function, D, can be compared with the theoretical one,  $D_{th}$ , as suggested by Johansson (Johansson and Gavel, 1994). D can be computed in two different ways:

- directly from its definition (Tatarski, 1961):

$$D(\mathbf{r}) = \overline{\left(u_x(\mathbf{s} + \mathbf{r}) - u_x(\mathbf{s})\right)^2};\tag{6}$$

where r is the separation vector, defined as the radial distance from the center of the grid;

- or, in a faster way, by means of the relation between the structure function and the correlation function, B, a statistical function that describes the mutual relation between the fluctuations of a physical phenomenon at different spatial positions. The procedure is the following:
  - 1. Compute the dataset's PSD,  $\Phi(k)$ , as:

$$\Phi(\mathbf{k}) = \overline{|\mathcal{F}(u(\mathbf{s}))^2|} = \frac{\mathcal{F}(u(\mathbf{s}))\mathcal{F}(u(\mathbf{s}))^*}{N};$$
(7)

where the symbol  $\mathcal{F}$  denotes the Discrete Fourier Transform (DFT).

2. Then the dataset's correlation function, B(r), is computed as the inverse Fourier transform of  $\Phi(k)$  (Wiener, 1930);

$$B(\mathbf{r}) = \mathcal{F}^{-1}(\Phi(\mathbf{k})). \tag{8}$$

3. Finally, the dataset's structure function can be computed as:

$$D(\mathbf{r}) = 2(B(\mathbf{0}) - B(\mathbf{r})); \tag{9}$$

where **0** is the null vector.

Once the dataset's structure function is computed using Eq. (9), the dataset's error can be quantified as:

$$\varepsilon(\mathbf{r}) = \left| \frac{D(\mathbf{r})}{D_{th}(\mathbf{r})} - 1 \right|. \tag{10}$$

For a more detailed explanation of the structure function and the correlation function, the interested reader is referred to (Tatarski, 1961).

#### 4 RPM's discretization errors

As an example of the RPM application, consider generating a 2-D wind field representing only the velocity component along the x-axis,  $u_x(s)$ . In this case, Eq. (4) becomes:

$$C_{xx}(\mathbf{k}) = \sqrt{\Phi_{xx}(\mathbf{k})}; \tag{11}$$

consequently,  $u_x(s)$  is:

100

$$u_x(\mathbf{s}) = \operatorname{Re}/\operatorname{Im}\left\{\mathcal{F}^{-1}\left(\mu(\mathbf{k})\frac{2\pi}{L_x L_y} N_x N_y \sqrt{\Phi_{xx}(\mathbf{k})}\right)\right\};\tag{12}$$

where  $L_i$  is the domain size along the *i*-th direction,  $N_i$  is the number of pixel along the *i*-th direction, and  $\mu(\mathbf{k})$  is a  $N_x \times N_y$  matrix of complex random variables with zero mean and unit variance. To test the RPM accuracy, a single wind field has been generated using Eq. (12) considering a von Kármán (VK) spectrum with a turbulence outer scale of  $L_0 = 756$  m, on a square grid of dimensions  $3L_0 \times 3L_0$ . The expected dataset' structure function is computed by setting  $\mu(\mathbf{k}) = 1$  in Eq. (12). Both the generated wind field, and its expected structure function are represented in Fig. (1). By analyzing the expected and theoretical structure functions, it is clear that the generated wind field does not properly represent the required statistics, underestimating the spectral power, and thus the wind speed variance, especially for larger values of the separation vector  $\mathbf{r}$ .

This error arises due to the fact that the PSD is not a bandwidth-limited function in the desired wavenumber domain, hence the implementation of the DFT generates a sampling error, as stated by the Nyquist-Shannon sampling theorem (Shannon, 1949). In fact, by using the DFT, the wavenumber domain on which the PSD is computed is determined by the grid's spatial resolution, as represented in Fig. (2), where it is clear how the spatial grid acts as a sort of bandpass filter, considering only part of the spectra. A possible way to circumvent the low wavenumbers' power underestimation is to increase the actual spatial grid size while maintaining the same spatial resolution in order to consider the lower wavenumbers. However, this approach depends on the considered PSD and on the grid shape, so it has to be optimized for each specific case. Moreover, for some applications (e.g., when employing this approach in a Monte-Carlo setup) this is not possible due to computational reasons, because of the

115

**Figure 1.** A wind field example generated by the RPM. Fig1a: a single wind field generated using the RPM. Fig1b: theoretical structure function in blue stars, dataset's expected structure function in dashed orange.

large memory size that the grid would end up having. Consequently, other methods that reduce this error calculation-wise have been developed.

For example, the *sub-harmonic method* (Sedmak, 1998) decreases the inaccuracies in the low-wavenumber region by replacing the single sample at the origin in the wavenumber domain by nine (or even more) sub-samples. These samples represent an equivalent length that is three times higher the length of a simple grid, allowing to sample the PSD at lower wavenumbers and enhancing the dataset accuracy at high spatial scales.

Another corrective method, proposed in (Xiang, 2014), implements a modal decomposition of the correlation function. The correlation function is pre-processed by extracting the piston and tilt components and a mask is applied. The tilt and piston components are then used to compute a tilt screen, that is added to the dataset generated using Eq. (12). However, the drawback of such methods is that they are not of general nature (i.e., they cannot be applied for all kinds of PSDs), hence it is needed to compute different weighting parameters to the PSD from spatial domain to spatial domain. Moreover, these methods can be applied only on square grids.

## 5 The correlation based method (CBM): generating the wind field from the correlation tensor

The here proposed CBM has been developed with the aim of eliminating the bandpass effect of the DFT arising while using the RPM method. To solve this problem, the relation between the correlation function and the PSD (i.e., the two quantities

**Figure 2.** The spatial grid acting as bandpass filter. The green rectangle represents the considered wavenumbers for a spatial domain of the order of three times the outer scale, in this case both a high and a low wavenumbers' power underestimation are expected. The red line represents the VK spectrum, and the black line represents the turbulence outer scale.

being a Fourier pair, as expressed in Eq. (8)) can be used. Indeed, by using the theoretical correlation function, computed on the spatial domain s, to generate the PSD used in the dataset's synthesis, the sampling error is eliminated. This solution, and how to generate a stochastic dataset from it, has been first investigated theoretically by (Cramer and Leadbetter, 1967), from which several methods, mainly used in the field of geostatistics, have followed. However some of these methods (Dietrich and Newsam, 1997; Wood and Chan, 1994) rely on the correlation matrix having a Toeplitz structure (i.e., a matrix in which each descending diagonal from left to right is constant). This is not always the case in wind field generation problems. Another proposed method (Pardo-Iguzquiza and Chica-Olmo, 1993) is very similar to the here proposed CBM, except that no arrangements of the Fourier coefficients on the wavenumber domain are needed in the latter one. The CBM can be described by the following steps:

1. **Compute the correlation function**: in the case of a homogeneous turbulent wind field, the correlation tensor can be computed as (Batchelor, 1953):

$$B_{pq}(\mathbf{r}) = \sigma^2 \left[ \frac{r_p r_q}{\mathbf{r}^2} f(\mathbf{r}) + \left( \delta_{pq} - \frac{r_p r_q}{\mathbf{r}^2} \right) g(\mathbf{r}) \right]; \tag{13}$$

where  $\sigma^2$  is the wind speed variance,  $f(\mathbf{r})$  is the longitudinal correlation function,  $g(\mathbf{r})$  is the lateral correlation function, and  $\delta_{pq}$  is the Dirac delta function.

140

2. Compute the PSD: the corresponding PSD,  $\Phi_{CBM}(\mathbf{k})$ , is the Fourier transform of the computed correlation function,  $B_{pq}(\mathbf{r})$ :

$$\Phi_{\text{CBM}}(\mathbf{k}) = \mathcal{F}(B_{pq}(\mathbf{r})); \tag{14}$$

the obtained  $\Phi_{\text{CBM}}$  is not the theoretical PSD, but it is the discrete Fourier pair of the theoretical correlation function  $B_{pq}(\mathbf{r})$  on the desired spatial domain. Note that  $\Phi_{\text{CBM}}(\mathbf{k})$  is not a tensor, as in (Mann, 1998), because it is the Fourier transform of the tensor component  $B_{pq}$ , and not of the whole tensor. This component takes already into account the longitudinal and lateral correlation, as expressed in Eq. (13).

3. Synthesize the dataset: the random dataset  $u_p(\mathbf{s})$  can be generated by taking the real or imaginary component of the DIFT of the square root of the PSD  $\Phi_{\text{CBM}}(\mathbf{k})$ , multiplied with a set of complex Gaussian random variables with zero mean and unity variance  $\mu(\mathbf{k})$ :

$$u_p(\mathbf{s}) = \operatorname{Re}/\operatorname{Im}\left\{\mathcal{F}^{-1}\left(\mu(\mathbf{k})\sqrt{\Phi_{\mathrm{CBM}}(\mathbf{k})N}\right)\right\}. \tag{15}$$

The main difference to the FIM lies in this last step. In this case there is no need to divide the wavenumber domain in different regions where the amplitude spectrum's coefficients (i.e., the square root of the PSD) are computed, but the PSD  $\Phi_{\rm CBM}(\mathbf{k})$  is directly used in the computation.

## 6 Validating the CBM at different spatial domains

As an example of the CBM application, a single component wind field has been generated on a 2-D domain using the same parameters as in Section 4. In the case of a 2-D domain and considering the velocity along the x-axis, Eq. (13) becomes:

$$B_{xx}(\mathbf{r}) = \sigma^2 \left[ \frac{r_x^2}{\mathbf{r}^2} f(\mathbf{r}) + \frac{r_y^2}{\mathbf{r}^2} g(\mathbf{r}) \right]; \tag{16}$$

and the generated wind field  $u_x(s)$  is:

$$u_x(\mathbf{s}) = \operatorname{Re}/\operatorname{Im}\left\{\mathcal{F}^{-1}\left(\mu(\mathbf{k})\sqrt{\mathcal{F}(B_{xx}(\mathbf{r}))N_xN_y}\right)\right\}. \tag{17}$$

The representation of  $u_x$  and its expected structure function are represented in Fig. (3). The expected structure function matches exactly the theoretical one, demonstrating that the synthesized dataset  $u_x(\mathbf{s})$  respects the required statistics.

In order to verify the actual generality of the method (i.e., it can be used without the need for any parameter optimization on any spatial domain), the expected structure function has been calculated for a wide range of spatial grid dimensions, starting from  $.01L_0$  up to  $10L_0$ . Similar to the function  $\Phi_{\rm FFT}$  in Xiang (Xiang, 2014), for very small spatial domains w.r.t. the turbulence outer scale  $L_0$ , the obtained PSD,  $\Phi_{\rm CBM}(\mathbf{k})$ , yields negative values. This is theoretically incorrect, because the PSD is defined as a positive function. The cause of the occurrence of these negative values is due to the implementation of the DFT on very small spatial domains. In fact, for very small spatial domains, the correlation function becomes almost flat, requiring

**Figure 3.** A wind field example generated by the CBM. Fig3a: a single wind field generated using the CBM. Fig3b: theoretical structure function in blue stars, dataset's expected structure function in dashed orange.

some of the DFT coefficients to be phase-shifted by a factor  $\pi$  (i.e., having negative-amplitude values) in order to obtain the desired shape in the spatial domain. This concept is represented in Fig. (4), where a single component wind field is generated on a 2-D,  $8 \times 8$  grid, of length  $0.1L_0$  and  $L_0 = 756$  m. By looking at Fig. (4), it is clear how using positive-amplitude DFT coefficients leads to an error in the calculation of the correlation function.

To tackle the issue regarding negative PSD values, it is possible to consider only the positive part of the PSD, and reduce the error by pre-compensating the correlation function, as proposed by (Xiang, 2014). Considering the example presented in Fig. (3), where the theoretical correlation function has been used without any compensation, this pre-compensation consists in updating the correlation function used in the PSD computation by subtracting to the theoretical correlation function,  $B_{xx}(r)$  defined in Eq. (16), the weighted error generated by neglecting the negative-amplitude wavenumbers of the PSD:

$$B_W(\mathbf{r}) = B_{xx}(\mathbf{r}) - W(\mathbf{r})\Delta_{\Phi}(\mathbf{r}); \tag{18}$$

where the error  $\Delta_{\Phi}(\mathbf{r})$  is computed as:

$$\Delta_{\Phi}(\mathbf{r}) = B_{xx}(\mathbf{r}) - \mathcal{F}\left[H\left(\Phi_{\text{CBM}}\left(\mathbf{k}\right)\right)\Phi_{\text{CBM}}\left(\mathbf{k}\right)\right];\tag{19}$$

where H() is the Heaviside function, and the weight W(r) has the shape of an elliptic super Gaussian window:

$$W(\mathbf{r}) = A e^{-C\left[\left(\frac{r_x}{E}\right)^g + \left(\frac{r_y}{F}\right)^g\right]};$$
 (20)

**Figure 4.** Negative DFT coefficients contribution. Fig4a: the theoretical correlation function in blue stars, the correlation function computed using Eq. (8) in orange, the correlation function computed using the absolute value of the PSD in dashed green. Fig4b: the different colors represent the contribution to the correlation function computation of different DFT coefficients. The contribution of the negative coefficients is represented by continuous lines, the contribution of the same coefficients but with a positive value in dashed lines. The difference between the sum of the two sets of lines gives the expected error in using the absolute value of the PSD.

**Table 1.** Compensation parameters

$$\begin{array}{lll} {\rm A} & \frac{65}{\left(\frac{L}{L_0}-0.7\right)1.2\sqrt{2\pi}}e^{-\frac{\left[\log\left(\frac{L}{L_0}-0.7\right)-1.7\right]^2}{2\times1.2^2}}-6.4\\ {\rm C} & 2\\ {\rm E} & L^2/530\\ {\rm F} & L^2/530\\ {\rm g} & 0.3 \end{array}$$

where g is the super Gaussian function exponent, and A,C,E,F are parameters that can be optimized depending on the PSD and the required dataset's accuracy. The parameters proposed for a square  $64 \times 64$  grid, of dimension  $L \times L$ , and a VK spectrum are reported in Table 1. These parameters were found by optimizing the dataset error for different spatial domains. By using the weighted CBM (WCBM) with these parameters, the obtained error is three to five times lower than using the CBM for very

© Author(s) 2025. CC BY 4.0 License.

small spatial domains w.r.t. the turbulence outer scale  $L_0$ . Finally, Eq. (17) becomes:

$$u_p(\mathbf{s}) = \operatorname{Re}\left\{\mathcal{F}^{-1}\left(\mu(\mathbf{k})\sqrt{\mathcal{F}(B_W(\mathbf{r}))N_xN_y}\right)\right\}. \tag{21}$$

## 7 Comparing the RPM and CBM methods

The errors of both methods have been compared on a wide range of spatial grid dimensions, starting from  $.01L_0$  up to  $10L_0$ . For each grid, the method error  $\Delta_i$  has been computed as:

$$\Delta_i = \max(\varepsilon(\mathbf{r}_i));$$
 (22)

where  $\varepsilon(\mathbf{r}_i)$  is computed using Eq. (10) for the separation vector  $\mathbf{r}_i$ , and the dataset's structure function  $D(\mathbf{r}_i)$  is computed using Eq. (9), in which the dataset's correlation function is computed as:

$$B(\mathbf{r}_i) = \overline{u_x(\mathbf{s})u_x(\mathbf{s} + \mathbf{r}_i)^*}.$$
 (23)

It has been decided to consider the maximum error value for each spatial domain for conservative reasons. The comparison between the two methods is represented in Fig. (5), where it is clear how the CBM over-performed the RPM of at least one order of magnitude, confirming that the method is a reliable solution in the synthesization of Gaussian processes without the need of any parameters optimization. For spatial domains greater than  $2.5L_0$ , a region where the generated wind fields are used to validate wind reconstruction algorithms' performances (Kiehn et al., 2022), the CBM can be declared as *exact in principle* (Wood and Chan, 1994), meaning that the method error is limited by the computer arithmetic inaccuracies. If a greater accuracy is needed at low spatial domains the WCBM can be used by implementing Eq. (21) (dashed green line in Fig. (5)). The proposed solution reduces the error up to two order of magnitudes w.r.t. the RPM.

## 8 Conclusions

200

205

In this publication a new generating method for synthesizing Gaussian and stationary phenomena has been presented. This new method allows to generate a dataset with an error of at least one order of magnitude less than the common used RPM. For spatial domains greater than  $2.5L_0$ , the CBM can be declared as exact in principle. Furthermore, being a general method, it can be used to synthesize any phenomenon considered Gaussian and stationary (e.g., index of refraction fluctuations, temperature fluctuations, homogeneous and isotropic turbulence) only by knowing the phenomenon's structure or correlation function. These can be obtained through analytical solutions (e.g., from the Kolmogorov cascade theory (Kolmogorov, 1991)) or from real measurement data, allowing to synthesize phenomena for which no analytical solutions exist (e.g., Clear Air Turbulence (CAT) events (Knox, 1997)). A further advantage of the method is that it allows, in a relatively simple way, the representation of anisotropies in the phenomenon of interest: these can be directly added to the theoretical correlation from which the PSD is computed (Pardo-Iguzquiza and Chica-Olmo, 1993). Finally, the method is valid also for non-symmetrical spatial domains, an interesting result that can be applied to highly customized wind speed dataset generation, as it will be described in a subsequent

**Figure 5.** Methods comparison. RPM error in blue, CBM error in orange, WCBM in dashed green. The x axis indicates the ratio between the considered spatial domain and the turbulence outer scale.

publication. Moreover, a routine to automatically compute the weighting function W(r) starting from the error  $\Delta_{\Phi}(r)$  is being developed. This would allow to achieve better performance on small spatial domains w.r.t. the turbulence outer scale  $L_0$ .

Author contributions. Problem statement: PV; conceptualization: all authors; methodology: MF; software: MF and DK; verification: DK; writing—review and editing: all authors. All authors have read and agreed to the published version of the manuscript.

Competing interests. The authors declared that there are any competing interests.

Disclaimer. Co-Funded by the European Union. Views and opinions expressed are however those of the authors only and do not necessarily reflect those of the European Union or Clean Aviation Joint Undertaking. Neither the European Union nor the granting authority can be held responsible for them.

https://doi.org/10.5194/wes-2025-221 Preprint. Discussion started: 18 November 2025 © Author(s) 2025. CC BY 4.0 License.

Acknowledgements. This study was performed within the framework of the UP Wing project. The project Ultra Performance Wing (UP Wing, project number: 101101974) is supported by the Clean Aviation Joint Undertaking and its members. The authors would like to acknowledge Dr. Tobias Bölle (DLR) and Dr. Lukas Bührend (DLR) for the insightful comments and fruitful discussion.

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
