# Peer review of "Generating high fidelity wind fields from the wind speed correlation tensor"

_Wind Energy Science, 2025_

## Author Comment (AC1)

- **As pointed out by the authors, the CBM method they propose does not seem to be that new. The novelty they added to the previous correlation-based methods is not clear at all. Could the authors give the reader more details about that ?**

Thanks for pointing it out. The novelty of the method stands in two features: w.r.t. the Fourier integral method (FIM) developed by (Pardo-Iguzquiza and Chica-Olmo, 1993) it does not need to divide the wavenumber domain in different regions where the amplitude spectrum's coefficients (i.e., the square root of the PSD) are computed, but the PSD is directly used in the computation, resulting in a faster computation having one less step. The novelty w.r.t. the methods computing a Toeplitz correlation matrix from the correlation function [Wood and Chan, 1994, Dietrich and Newsam, 1993] is that the CBM allows to synthesize the dataset directly from the wind correlation function. The novelty of this method w.r.t. the RPM lies in its performance, especially for spatial domains larger than $2.5L_0$.
*Action: Explained with more details the novelty of the method in section 5 and 7.*

- **Have the authors thought about addressing the more challenging issue of generating non-isotropic, non-homogeneous synthetic turbulence ?**

Yes, at the moment it is being considered to use the proposed method to synthesize anisotropic wind fields based on datasets measured from aircraft (5-hole-probe) or from ground. Regarding non-homogeneous turbulence it is more complex, being the Fourier-Stieltjes representation of the wind valid only for a homogeneous random field. However, a possible method to overcome this limitation was proposed in [Wilson, 1997]. Thanks for pointing out this was not clearly explained.
*Action: Added the possible future investigation of anisotropic wind using the CBM in the conclusions.*

- **L124: the authors point out the limitation of previous correlation based methods (Dietrich and Newsam, 1997; Wood and Chan, 1994) due to the fact they "on the correlation matrix having a Toeplitz structure" To the reviewer's limited knowledge, the Toeplitz structure of the correlation matrix is directly linked to the fact that one considers homogeneous (or stationary) directions, which is mandatory to be able to use Fourier transforms. As the authors have designed their method to generate isotropic homogeneous turbulence, their method has the same requirement: a correlation matrix with a Toeplitz structure. Could the authors comment on that please ?**

Indeed this was a misunderstanding by the authors. In misunderstanding how the correlation matrix is defined in the papers by [Dietrich and Newsam, 1993] and [Wood and Chan, 1994], it was thought it is the same

correlation function as defined in [Batchelor, 1953], the latter not having a Toeplitz structure on a 2-D domain. Even if it was a trivial error, this does not affect the concept behind the CBM.
*Action: the error has been corrected.*

- **L125: "This is not always the case in wind field generation problems." could the authors give some example of cases were correlation matrices are not of Toeplitz type but which their method could handle anyway ?**

  (Related to above). As explained in the previous answer, this error was due to misunderstanding the correlation function definition in the papers by [Dietrich and Newsam, 1993] and [Wood and Chan, 1994]. Assuming homogeneous wind, following [Zimmermann, 1989] the correlation matrix, as defined in the method described in [Dietrich and Newsam, 1993] and [Wood and Chan, 1994], is always block Toeplitz.
  *Action: the error has been corrected.*

- **Fig 1: how the ideal structure function is calculated ?**

  Thanks for pointing this out, it was not clear. The theoretical structure function is computed using the following equation [Tatarski, 1961]:

  $$D(\vec{r}) = 2(B(\vec{0}) - B(\vec{r})) \, ; \tag{1}$$

  where in the case of the velocity component along the x-axis in a 2-D domain the correlation function $B(\vec{r})$ is [Batchelor, 1953]:

  $$B(\vec{r}) = \sigma^2 \left[ \frac{r_x^2}{\vec{r}^2} f(\vec{r}) + \frac{r_y^2}{\vec{r}^2} g(\vec{r}) \right] \, ; \tag{2}$$

  where $\sigma^2$ is the wind speed variance, and $f(\vec{r})$ and $g(\vec{r})$ are the longitudinal and lateral correlation functions respectively. Assuming a von Kármán spectrum, these can be expressed as [Wilson, 1998]:

  $$f(\vec{r}) = \frac{2}{\Gamma(1/3)} \left( \frac{\vec{r}}{2L_0} \right)^{1/3} K_{1/3} \left( \frac{\vec{r}}{L_0} \right) \, ; \tag{3}$$

  $$g(\vec{r}) = \frac{2}{\Gamma(1/3)} \left( \frac{\vec{r}}{2L_0} \right)^{1/3} \left[ K_{1/3} \left( \frac{\vec{r}}{L_0} \right) - \frac{\vec{r}}{2L_0} K_{2/3} \left( \frac{\vec{r}}{L_0} \right) \right] \, ; \tag{4}$$

  where $K_\nu$ is the modified Bessel function of the second kind of order $\nu$.
  *Action: These equations have been added in section 4.*

- **For the RPM, the authors prescribe a von Karman spectrum (I presume it is one-dimensional), whereas for the CBM, they prescribe the full 3D tensor $B_{pq}$. What is the impact of these difference ?**

  No, in both cases the wind correlation tensor is used. Thanks for pointing out that this was unclear. This is clarified by adding the equations from

the previous answer.

*Action: Added the theoretical structure function computation from the wind correlation tensor.*

- **L156 and subsequent: the description and the origin of the problem related to PSD with negative values is not clear at all. Could the author elaborate on that ?**

Thank for pointing it out. The authors tried to divide the explanation in two different parts. First, why negative values are appearing in the PSD? Considering the extreme case in which the correlation function is a circular flat surface, its Fourier transform would be a *jinc* function [Goodman, 1996]. So the appearance of negative values in $\Phi_{CBM}\left(\vec{k}\right)$ is expected for very small spatial domains. Second, it is tried to explain the error that would be committed if positive values of the PSD, computed as the DFT of the correlation function, are considered. The obtained error is represented in Fig.(1), where a single component wind field is generated on a 2-D, $8 \times 8$ grid, of length $0.1L_0$, and $L_0 = 756$ m. It is clear how using positive-amplitude DFT coefficients leads to an error in the calculation of the correlation function.

*Action: Added this explanation in section 6.*

- **Could the authors show the spectra of the velocity field obtained using the CBM and compare it to the expected spectrum and that used with the RPM ?**

It was thought to add also the PSD plots. But to avoid biases generated by computing the dataset's PSD as the modulus squared of the DFT, or underestimation of the dataset's power due to windowing [Thomson, 1982], it was considered more appropriate to compare the synthesized datasets' correlation functions, computed as:

$$B(\vec{r}_i) = \overline{u_x(\vec{s})u_x(\vec{s} + \vec{r}_i)^*} \, . \tag{5}$$

For the Wiener–Khinchin theorem, the obtained correlation function is the Fourier transform of the PSD. So in the case the dataset's correlation function error w.r.t. the theoretical one is low, this applies also to the dataset's PSD.

*Action: Added this explanation in section 7.*

- **Fig 5: could the author explain why the error of the RPM remains independent of the size of the spatial domain ? It seems to contradict the point they made in the last paragraph of page 4 (line 98) stating that the grid domain acts as a bandpass filter. One would expect the performance to increase as the bandwidth of the filter increases (with domain size).**

The apparent independence of the RPM error from the considered spatial domain is due to how the error is computed in the publication. By increasing the considered spatial domains the error tends to increase. This

[Figure]

Figure 1: Negative DFT coefficients contribution. Fig4a: the theoretical correlation function in blue stars, the correlation function computed assuming negative values in the PSD in orange, the correlation function computed using the absolute value of the PSD in dashed green. Fig4b: the different colors represent the contribution to the correlation function computation of different DFT coefficients. The contribution of the negative coefficients is represented by continuous lines, the contribution of the same coefficients but with a positive value in dashed lines. The difference between the sum of the two sets of lines gives the expected error in using the absolute value of the PSD.

is expected in the case the error is considered as the maximum of the function:

$$\varepsilon(\vec{r}) = \left| \frac{D(\vec{r})}{D_{th}(\vec{r})} - 1 \right| . \tag{6}$$

Indeed, by increasing the spatial domain, keeping the same number of pixels, more low frequencies are considered, while less high frequencies are considered. This leads to a higher error at low separation distances, where the structure function is very low, so it is expected the error to increase. Two other possible way of quantifying the error are presented below. Considering the standard deviation of the error $\varepsilon$, the obtained errors for different spatial domains are represented in Fig.(2).

While considering the absolute error:

$$\epsilon_A(\vec{r}) = |D(\vec{r}) - D_{th}(\vec{r})| ; \tag{7}$$

the obtained errors for different spatial domains are represented in Fig.(3). The purpose of this analysis was to underline the better performances of the CBM w.r.t. the RPM, so the authors think that the maximum of the

[Figure]

Figure 2: Methods comparison considering the $\text{std}(\varepsilon)$. RPM error in blue, CBM error in orange. The x axis indicates the ratio between the considered spatial domain and the turbulence outer scale.

[Figure]

Figure 3: Methods comparison considering the absolute error $\epsilon$. RPM error in blue, CBM error in orange. The x axis indicates the ratio between the considered spatial domain and the turbulence outer scale.

error is a good parameter to quantify it. Moreover, the WCBM has been optimized using $\max(\varepsilon)$ as merit function.

*Action: Increased the considered spatial domains in the RPM-CBM com-*

*parisons figure to show the increase of the error.*

- **It would interesting to show the structure functions computed for the various domain size for both methods RPM and CBM.**

  Yes, indeed in the text only two structure functions are plotted (one for the RPM and one for the CBM). Nine different structure functions, computed for different spatial domains, are represented in Fig.(4).

  However the authors think that the error of the RPM w.r.t. the CBM can be represented by the structure functions already presented in the paper, to not overfill the papers with graphs. If the reviewer(s) favor the display of these examples, the authors surely might include this set.

The authors would like to thank the reviewer for her/his time and effort. All the raised points were very helpful in pointing out the parts not clear, and in tracking down the remaining mishaps.

**References**

A. T. A Wood and G. Chan. Simulation of stationary gaussian processes in [0, 1]. *Journal of Computational and Graphical Statistics.*, 3(4), 1994.

C. R. Dietrich and G. N. Newsam. A fast and exact method for multidimensional Gaussian stochastic simulations. *Water Resources Research*, 29(8), 1993. doi: doi.org/10.1029/93WR01070.

D. K. Wilson. Three-dimensional correlation and spectral functions for turbulent velocities in homogeneous and surface-blocked boundary layers. *Army Research Laboratory: Adelphi, MD, USA*, 7 1997.

G. K. Batchelor. *The theory of homogeneous turbulence.* Cambridge University Press, 1953.

D. L. Zimmermann. Computationally exploitable structure of covariance matrices and generalized convariance matrices in spatial models. *Journal of Statistical Computation and Simulation*, 32, 1989. doi: 10.1080/00949658908811149.

V. I. Tatarski. *Wave Propagation in a Turbulent Medium.* McGraw-Hill Book Company, Inc., 1961.

D. K. Wilson. Turbulence Models and the Synthesis of Random Fields for Acoustic Wave Propagation Calculations. *Army Research Laboratory: Adelphi, MD, USA*, 7 1998.

J. W. Goodman. *Introduction to Fourier Optics.* McGraw-Hill, 1996.

D. J. Thomson. Spectrum estimation and harmonic analysis. *Proceedings of the IEEE*, 70(9), 9 1982. doi: 10.1109/PROC.1982.12433.

[Figure]

Figure 4: Different structure functions, computed for different spatial domains. On top of each subplot is written the considered spatial domain. In red stars the ideal structure function, in blue line the structure function computed from a RPM generated dataset, in orange line the structure function computed from a CBM generated dataset, in red dots the structure function computed from a WCBM generated dataset. For all the considered spatial domains, $L_0 = 756\,\mathrm{m}$ and $\sigma^2 = 1\,\mathrm{m}^2\,/\,\mathrm{s}^2$.

---

## Author Comment (AC2)

- **This paper discusses the generation of homogeneous, isotropic wind fields. The aim is to improve the quality of reconstructed fields by using knowledge of the correlation tensor. The paper is technically interesting and well written. It provides a useful overview of previous work, but the novelty of this work could be highlighted more effectively.**

  Thanks for pointing it out.

  *Action: The novelty of the method has been explained in more details in sections 5 and 7.*

- **My main criticism, which is left, concerns the discussion of the possible application of this work to wind energy systems, i.e. generating real wind fields. How can the IEC-wind field generators be improved? A comment on how to extend the proposed method to simulate full three-component wind fields would be helpful.**

  Thanks for the comment. At the moment it is possible to use the CBM considering only an homogeneous and isotropic wind field, described by the correlation function defined in [Batchelor, 1953]. In a subsequent publication, it is intended to extend the CBM modeling capabilities adding the Kaimal [Kaimal et al., 1972] and the Mann [Mann, 1994] PSDs. To do so the respective correlation functions shall be computed (i.e., computing the inverse Fourier transform of the PSDs), and then used as the wind field required correlation, substituting the homogeneous wind correlation function (i.e., Eq.(16) in the proposed paper). Even if in the example in the proposed paper covers a 2-D wind field, it is already possible to synthesize 3-D wind fields considering the correlation function for a 3-D domain:

  $$B(\vec{r}) = \sigma^2 \left[ \frac{r_x^2}{\vec{r}^2} f(\vec{r}) + \frac{r_y^2 + r_z^2}{\vec{r}^2} g(\vec{r}) \right] . \tag{1}$$

  *Action: The intention to add the Kaimal and Mann PSDs in a future work has been added in the conclusions.*

- **Coming to real wind fields. Some remarks o the following questions would be of value: How can non-homogeneous situations be handled? A big problem will be non-stationarity. Which quantities should be measured in the field? I also question whether the increased accuracy shown is really valuable for real wind fields. I would imagine that the intrinsic errors of the corrections of wind field measurements are so high than the proposed increase in accuracy. To discuss this frankly does not lower the quality of the work but is of interest for application.**

  Thank you for pointing it out. In fact, this was not mentioned at all in the proposed paper. The idea behind the CBM is to have a method that

does not introduce computational errors and then refine the wind field synthesis using field data (e.g., measured using anemometers, five-hole probes). From these data a more realistic wind correlation function can be inferred, to then be used in the synthesis of wind fields. Regarding modeling non-stationary wind, the method proposed in [Wilson, 1997] is being investigated.

*Action: The comment on the CBM error in section 7 has been extended adding this answer.*

- **Another question is whether the authors have any ideas about how to include higher-order correlations/statistics. For example, does the reconstructed wind field reproduce the well-known skewness scaling of ideal turbulence?**

  Thank you very much for this comment. In the paper the authors focused only on a second-order statistics, assuming a Gaussian distribution of the wind field. However, it would be possible to add non-Gaussian features to the synthesized wind field following [Friedrich et al., 2022]. This is being considered for future work.

  *Action: The possibility to add non-Gaussian features in a future work has been added in the conclusions.*

The authors would like to thank the reviewer for her/his time and effort. All the raised points were very helpful in pointing out the parts not clear and to clarify how the method will evolve in future works.

**References**

G. K. Batchelor. *The theory of homogeneous turbulence.* Cambridge University Press, 1953.

J. C. Kaimal, J. C. Wyngaard, Y. Izumi, and O. R. Coté. Spectral characteristics of surface-layer turbulence. *Quarterly Journal of the Royal Meteorological Society*, 98, 1972. doi: 10.1002/QJ.49709841707.

J. Mann. The spatial structure of neutral atmospheric surface-layer turbulence. *J. Fluid Mech.*, 273, 1994. doi: 10.1017/S0022112094001886.

D. K. Wilson. Three-dimensional correlation and spectral functions for turbulent velocities in homogeneous and surface-blocked boundary layers. *Army Research Laboratory: Adelphi, MD, USA*, 7 1997.

J. Friedrich, D. Moreno, M. Sinhuber, M. Wätcher, and J. Peinke. Superstatistical wind fields from pointwise atmospheric turbulence measurements. *PRX ENERGY*, 1, 2022. doi: 10.48550/arXiv.2203.16948.